# 🛋 CRAFT: A Benchmark for Causal Reasoning About Forces and inTeractions

**Tayfun Ates**[1,*]
tates@hacettepe.edu.tr

**M. Samil Atesoglu**[1,*]
matesoglu@hacettepe.edu.tr

**Cagatay Yigit**[1,*]
cyigit@hacettepe.edu.tr

**Ilker Kesen**[2]
ikesen16@ku.edu.tr

**Mert Kobas**[3]
mkobas18@ku.edu.tr

**Erkut Erdem**[1]
erkut@hacettepe.edu.tr

**Aykut Erdem**[2]
aerdem@ku.edu.tr

**Tilbe Goksun**[3]
tgoksun@ku.edu.tr

**Deniz Yuret**[2]
dyuret@ku.edu.tr

[1] Hacettepe University Computer Vision Lab    [2] Koç University Is Bank AI Center
[3] Koç University Language and Cognition Lab
https://sites.google.com/view/craft-benchmark

## Abstract

Humans are able to perceive, understand and reason about physical events. Developing models with similar physical understanding capabilities is a long standing goal of artificial intelligence. As a step towards this goal, in this work, we introduce CRAFT, a new visual question answering dataset that requires causal reasoning about physical forces and object interactions. It contains 58K video and question pairs that are generated from 10K videos from 20 different virtual environments, containing various objects in motion that interact with each other and the scene. Two question categories from CRAFT include previously studied *descriptive* and *counterfactual* questions. Besides, inspired by the theories of force dynamics in cognitive linguistics, we introduce new question categories that involve understanding the interactions of objects through the notions of *cause*, *enable*, and *prevent*. Our results demonstrate that even though these tasks seem to be simple and intuitive for humans, the evaluated baseline models, including existing state-of-the-art methods, do not yet deal with the challenges posed in our benchmark dataset.

## 1 Introduction

The cognitive capabilities of humans to understand and make approximate predictions about physical objects and their interactions are known as *intuitive physics* [1]. Cognitive scientists have extensively studied the factors that affect physical reasoning in infants or adults [2–5]. Some of these abilities have also been studied for other animals such as chicks (Gallus gallus) [6]. Recent advances in machine learning have enabled computers to understand what type of object is present in a specified image (*classification*), which bounding box best wraps that object (*detection*), what its exact boundaries are (*segmentation*). Although these artificial vision systems have shown astounding progress in the past decade, there are some areas in which these systems are still significantly below human performance. One such area includes the capability of humans to reason about physical actions of objects by observing their environment. In this line of work, cognitive and computer scientists are

---

*indicates equal contributions.

Submitted to the 35th Conference on Neural Information Processing Systems (NeurIPS 2021) Track on Datasets and Benchmarks. Do not distribute.

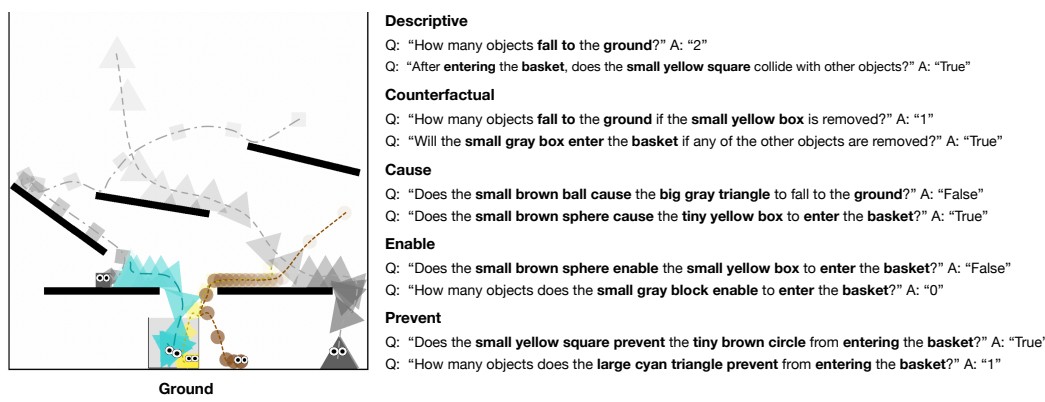

**Descriptive**

Q: "How many objects **fall to** the **ground**?" A: "2"

Q: "After **entering** the **basket**, does the **small yellow square** collide with other objects?" A: "True"

**Counterfactual**

Q: "How many objects **fall to** the **ground** if the **small yellow box** is removed?" A: "1"

Q: "Will the **small gray box enter** the **basket** if any of the other objects are removed?" A: "True"

**Cause**

Q: "Does the **small brown ball cause** the **big gray triangle** to fall to the **ground**?" A: "False"

Q: "Does the **small brown sphere cause** the **tiny yellow box** to **enter** the **basket**?" A: "True"

**Enable**

Q: "Does the **small brown sphere enable** the **small yellow box** to **enter** the **basket**?" A: "False"

Q: "How many objects does the **small gray block enable** to **enter** the **basket**?" A: "0"

**Prevent**

Q: "Does the **small yellow square prevent** the **tiny brown circle** from **entering** the **basket**?" A: "True"

Q: "How many objects does the **large cyan triangle prevent** from **entering** the **basket**?" A: "1"

Figure 1: **Example CRAFT questions generated for a sample scene.** There are 48 different tasks divided into 5 distinct categories for 20 different scenes. Besides having tasks questioning descriptive properties, possibly needing temporal reasoning, CRAFT introduces challenges including more complex tasks requiring single or multiple counterfactual analysis or understanding object intentions for deep causal reasoning.

working together to bring similar capabilities to artificially intelligent systems so that they acquire similar intuitions and better understand their surroundings.

Importantly, improving physical reasoning capabilities can make agents better anticipate the results of their actions in their physical environments. They can gain the ability to consider counterfactual actions without actually performing them. They can estimate what will happen if they perform a specific action. One of the recent examples in this direction is the Jenga-playing robot [7]. We believe intuitive physics is an essential ability to develop machines that are safe to interact with humans.

In this work, our main aim is to judge how well the existing neural models understand and reason about physical relationships between dynamic objects in a scene. We propose a new visual question answering task, named CRAFT (Causal Reasoning About Forces and inTeractions), which requires understanding complex physical reasoning to be able to score high. CRAFT is designed to be complex for artificial models and simple for humans. Our dataset contains virtually generated videos of 2-dimensional scenes with accompanying questions. Its most prominent properties are that it contains video clips with complex physical interactions between objects and questions that test strong reasoning capabilities. For example, answering the questions needs understanding what is being asked, and requires detecting objects, tracking their states in relation to other objects, which in turn can be attributed to causing, enabling or preventing certain events. Moving beyond simple causal relations, enable and prevent categories refer to interactions between multiple forces. Distinct causal verbs are mapped onto these three classes of causal events. Moreover, there are also counterfactual questions about understanding what would have happened after an intervention, i.e. a slight change in the environment [8]. Figure 1 shows sample questions from CRAFT from 5 different categories, which are explained in detail in the subsequent sections, for a single simulation[2].

Our main contribution is the creation of a novel dataset that uses language and vision to test spatiotemporal reasoning on complex physical systems. In addition, we experiment with some simple and strong baselines and demonstrate that they are insufficient to handle the challenges CRAFT introduces. We hope that our work will lead to the generation of better systems on the path of approaching the level of human intelligence for physical reasoning.

## 2 Related Work

**Visual Question Answering.** Existing visual question answering (VQA) datasets can be categorized along two dimensions. The first dimension is the type of visual data, which include either real world images [9–13] or videos [14, 15], or synthetically created content [16–18]. The second is

---

[2]More examples from CRAFT can be found in Appendix A.3 and also on the project website, located at http://sites.google.com/view/craft-benchmark.

at how the questions and answers are collected, which are usually done via crowdsourcing [9, 11] or by automatic means [10, 19, 16]. An important challenge for creating a good VQA dataset lies in minimizing the dataset bias. A model may exploit such biases and cheat the task by learning some shortcuts. In our work, we generate questions about simulated scenes using a pre-defined set of templates by considering some heuristics to eliminate strong biases. As compared to the existing VQA datasets, our CRAFT dataset is specifically designed to test the agents' understanding of dynamic state changes of the objects in a scene. Although some existing VQA datasets question temporal reasoning [15, 20–22], they do not require the models to have a deep understanding of intuitive physics to answer the questions, the only exceptions being TIWIQ [23], CLEVRER [18], and CLEVR_HYP [24] datasets. In these datasets, there exist some hypothetical questions that require mental simulations about the consequences of performing certain actions or the lack of specific actions or objects. These datasets have received interest in the community to develop reasoning models with physical understanding capabilities, e.g., the neural-symbolic approaches proposed in [25, 26]. CRAFT shares a similar design goal with these aforementioned TIWIQ, CLEVRER, and CLEVR_HYP datasets – however the scenes in our benchmark are more complex, as explained later.

**Intuitive Physics in Cognitive Science.** Common sense is considered as the collection of human reasoning abilities to perceive, understand and judge everyday situations. Intuitive physics, an important part of commonsense knowledge, is related to people's perceptions of changes in physical world and their own understanding of how physical phenomena works [27]. Different theories have been proposed by cognitive scientists to model how humans learn, experience, and perform physical reasoning for certain events. Some of them are mental model theory [28], causal model theory [29], and force dynamics theory [30], which try to represent a variety of causal relationships such as cause, enable, and prevent between two main entities, an affector and a patient (the object the affector acts on). To our knowledge, our work is the first attempt at integrating these complex causal relationships in a VQA setup for machine learning models to improve their physical reasoning capabilities.

**Intuitive Physics in Artificial Intelligence.** In recent years, there has been a growing interest within the AI community in developing models that have reasoning about intuitive physics. For instance, some researchers have explored the problem of predicting whether a set of objects are in stable configuration or not [31] or if not where they fall [32]. Others have tried to estimate a motion trajectory of a query object under different forces [31] or developed methods to build a stack configuration of the objects from scratch through a planning algorithm [33]. [34] suggested to represent rigid bodies, fluids, and deformable objects as a collection of particles and used this representation to learn how to manipulate them. Very recently, Bakhtin et al. [35] and Allen et al. [36] created the PHYRE and the Tools benchmarks, respectively, which both include different types of 2D-environments. An agent must reason about the scene and predict the outcomes of possible actions in order to solve the task associated with the environment. CoPhy [37] is another recent benchmark, which deals with physical reasoning prediction about counterfactual interventions. Although these works involve complicated physical reasoning tasks, the language component is largely missing. As mentioned earlier, Wagner et al. [23], Yi et al. [18] and [24] created VQA datasets for intuitive physics, but they lack visual variations unlike PHYRE and Tools. In that sense, our CRAFT dataset combines the best of both worlds. Moreover, in addition to the two types of questions investigated in CLEVRER [18], namely *descriptive* and *counterfactual*, CRAFT also involves questions that need reasoning about the concepts like *cause*, *enable*, and *prevent*. To succeed in these tasks, the machine reasoning models need to learn the semantics of each verb category that specifies different kinds of interactions between objects, i.e. in a way, need have a kind of commonsense knowledge.

## 3   The CRAFT Dataset

CRAFT is built to evaluate temporal and causal reasoning capabilities of existing algorithms on video clips of 2D simulations and related questions. The dataset has approximately 57K question and video pairs, which are created from 10K videos. It is split into train, validation, and test sets with a 60:20:20 ratio per video basis, meaning that video clips in the training set are not seen in the validation or test set. Moreover, we have two different settings, an *easy setting* and a *hard setting*. They differ from each other in the way how the test split is chosen. In the hard setting, we deliberately use scene types that are not seen during training in picking the video and question pairs. The easy setting does not have this constraint. In the easy setting, there are 35K, 12K, and 11K question and video pairs in the train, validation and test splits, whereas in the hard setting these numbers are 35K,

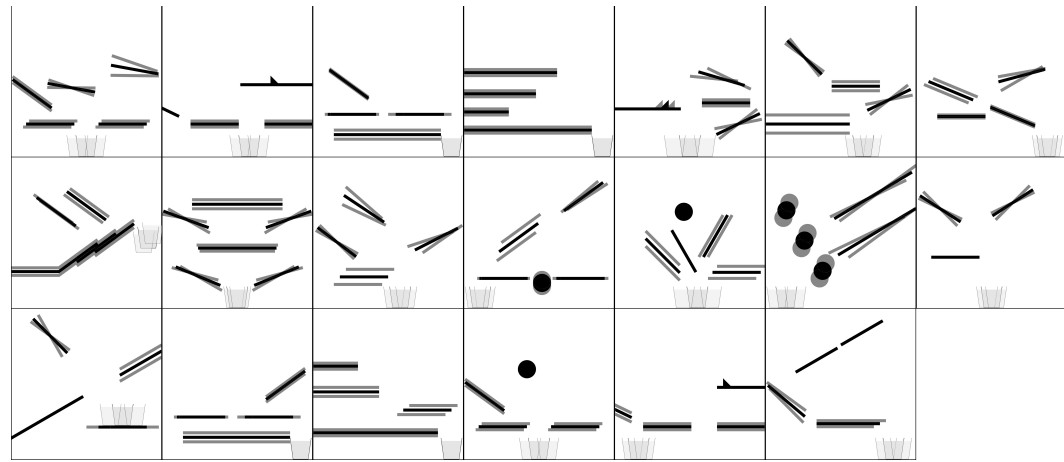

Figure 2: **Random configurations of static scene element properties for each scene.** The opaque regions show the mean value for that element, whereas the overlayed regions show the extreme values. Although these changes may seem subtle, they provide a wide variety in terms of scene dynamics.

11K and 12K, respectively. We provide an example set of questions from CRAFT in Figure 1. In what follows, we are going to mention how we generate visual scenes, which types of objects and events exist in videos and questions, how we represent our simulations, how we define the tasks and accordingly generate the questions, and finally, how we reduce the biases that might easily emerge in visual question answering datasets.

**Video Generation.** We use Box2D physics simulator [38] to create our virtual scenes. There are 20 distinct scene layouts from which 10 seconds of video clips are collected with a spatial resolution of $256 \times 256$ pixels. Besides generating original simulation video, CRAFT scripts also generate variation videos by removing each object of the same video from the scene. These variation videos help question generation script to provide answer for certain types of questions, as explained later.

**Objects.** Each scene is composed of both *static scene elements* and *dynamic objects*, containing variable number of and different type of these elements and objects. There are 7 static scene elements (*ramp*, *platform*, *button*, *basket*, *left wall*, *right wall*, *ground*). These elements are all drawn in **black** color in order to differentiate them from the dynamic objects. Their attributes such as position or orientation are decided at the beginning of a simulation and then they are kept fixed throughout the video sequence. The values of these attributes are assigned randomly from sets of different intervals which are predefined for each type of scene as in Figure 2. The set of the dynamic objects contains 3 shapes (*cube*, *triangle*, *circle*), 2 sizes (*small*, *large*), and 8 colors (*gray*, *red*, *blue*, *green*, *brown*, *purple*, *cyan*, *yellow*). Attributes of dynamic objects, on the other hand, are in continuous change throughout the sequence due to the gravity or the interactions that they are subject to, until they rest.

**Events.** To formally represent the dynamical interactions in the simulations, we extract different types of events. These events are *Start*, *End*, *Collision*, *Touch Start*, *Touch End*, and *Enter Basket*. *Start* and *End* events represent the start and the end of the simulations, respectively. Although we mainly question *Collision* events in our tasks, we want models to understand the difference between a collision and rolling on a ramp or a platform or two objects moving together. Therefore, we also extract *Touch Start*, *Touch End* events. Finally, *Enter Basket* event is triggered if the object enters the basket in the scene. All events happening a simulation are represented as a causal graph, which is also key for the question generator to extract causal relationships in an easy manner. Causal graph is a directed graph where events are represented as nodes. Each edge represents a cause relation where the source event is considered as the cause of target event because of the shared objects between them. We demonstrate the causal graph of a sample simulation in Figure 3.

**Simulation Representation.** A simulation instance is represented by 3 different data structures, which are *the initial state of the scene*, *the final state of the scene*, and *the causal graph of extracted events*. The inial and final state of a scene refers to the information regarding the objects' static and dynamic attributes such as color, position, shape, and velocity. at the start or at the end of the simulation, respectively. The final state is important as it bears causal relationships between the

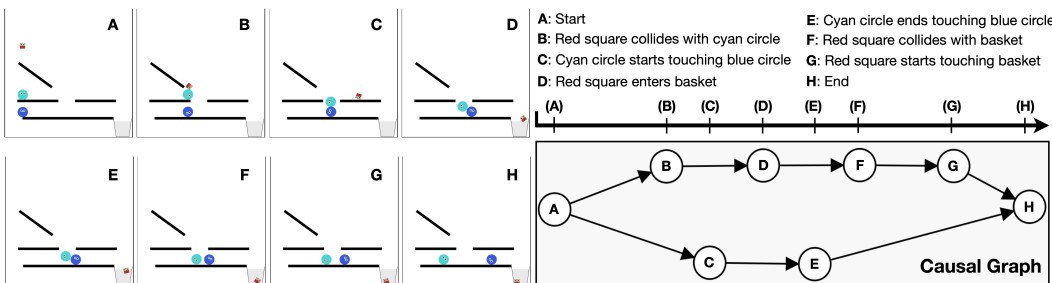

Figure 3: **A simple causal graph.** The causal graph shows the graphical representations of the events that occur in a simulation. For the sake of simplicity, here we only include the interactions between the dynamic objects and the basket and moreover the scene is uncomplicated that there is no intermediate branching in the causal graph.

events of a simulation. Together these information sources have sufficient information to find the correct answers to CRAFT questions. Our simulation system also allows us to generate scene graphs like the ones used in CLEVR [16], though we have not investigated it yet, which might be used for spatial reasoning.

**Question Generation.** Each CRAFT question is represented with a functional program as in CLEVR. We use a different set of functional modules for our programs extending the CLEVR approach. For example, our module set includes, but is not limited to functions which can filter events such as *Enter Basket* and *Collision*, and functions which can filter objects based on whether they are stationary at the start or the end of the video. List of our functional modules and some example programs are provided in Appendices A.1 and A.2 in the supplementary material, respectively. Moreover, we use different sets of word synonyms and allow question text to be paraphrased for language variety similar to CLEVR. Our preliminary analysis reveals that human performances in some questions are very poor. When investigated, we figure out that these questions seem to be counter-intuitive to humans. Humans do not accurately reason about the objects for some counterfactual cases as subtle changes in the scenes result in very different outcomes. Hence, while finalizing our dataset, we apply minor random perturbations to each dynamic object in a video to verify whether the same answer is obtained for all such cases, and exclude those questions that do not pass this verification step.

**Question Types.** CRAFT has 48 different question types under 5 different categories, namely *Descriptive, Counterfactual, Enable, Cause, Prevent*. Among these, *Descriptive* questions mainly require extracting the attributes of objects, but some of them, especially those involving counting, need temporal analysis as well. Our dataset extends CLEVRER by Yi et al. [18] with different types of events and multiple environments. *Counterfactual* questions require understanding what would happen if one of the objects was removed from the scene. Exclusive to CRAFT, some *Counterfactual* questions (*"Will the small gray circle enter the basket if any of the other objects are removed?"*) require multiple counterfactual simulations to be explored. As an extension to *Counterfactual* questions, *Enable, Cause, Prevent* questions require grasping what is happening inside both the original video and the counterfactual video. In other words, models must infer whether an object is causing or enabling an event or preventing it by comparing the input video and the counterfactual video that should be simulated somehow. In the question text, the affector and the patient objects are explicitly specified. Some questions even include multiple patients.

In order to have a better understanding of the differences between *Enable*, *Cause*, and *Prevent* questions, one should understand the *intention* of the objects. We identify the intention in a simulation by examining the initial linear velocity of the corresponding object. If the magnitude of the velocity is greater than zero, then the object is intended to perform the task specified in the question text, such as entering the basket or colliding with the ground. If the magnitude of the velocity is zero, then it is assumed that the object has no such intention – even if there is an external force such as gravity, upon it at the beginning of the simulation. Therefore, an affector can only enable a patient to complete the task if the patient is originally intended to do it but fails without the affector. Similarly, an affector can only cause a patient to do the task if the patient is not intended to execute it. Moreover, an affector can only prevent a patient from completing the task if the patient is intended to do it and succeeds without the affector.

**Variations in Natural Language.** In datasets that involve a natural language component, it is crucial to have language variety. To improve this property, CRAFT data generation scripts for questions, first allow multiple paraphrased versions of the same text to be generated to represent the same task. For a question sample, a paraphrased version of the corresponding task is chosen randomly by filling the object templates. Second, CRAFT enables synonyms of certain words to be integrated. We choose a base word and create its synonyms inside the CRAFT context. Similar to question paraphrases, the base word is replaced by a synonym randomly at run-time. All synonyms including the base word have equal chance to be included in the question text. This replacement is handled by word suffixes and verb conjugations by preserving English grammar.

**Bias Reduction.** CRAFT contains simulations from different scenes increasing the variety in the visual domain as well. This variety also makes reducing the dataset biases difficult because of the multiplicity in the number of the domains (textual and visual). Our data generation process enforces different simulation and task pairs to have uniform answer distributions while trying to keep overall answer distribution as uniform as possible.

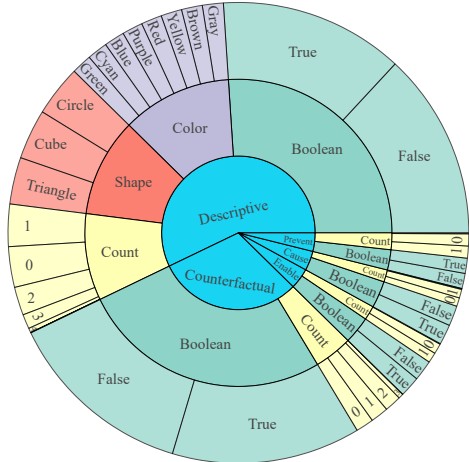

Figure 4: **Distribution of question types and answers in CRAFT.** Innermost layer represents the distribution of the questions for different task categories. Middle layer illustrates the distribution of the answer types for each task category. Outermost layer represents the distribution of answers for each answer type.

Here, our aim is to make it harder for the models to find simple shortcuts by predicting the task identifier, the simulation identifier, or both, instead of understanding the scene dynamics and the question. Figure 4 shows the answer distributions for the question categories in CRAFT.

# 4 Experimental Analysis

In this section, we evaluate the performances of a wide range of baseline models on our CRAFT dataset. We also analyze how these performances relate with that of humans in understanding physical interactions between the objects and the environment.

## 4.1 Baselines

In our experiments, we consider several baseline models including the state-of-the-art visual reasoning approaches. In the following, we give details of these models. In particular, five of these models are text-only baselines which only read the question and give an answer without looking any of the video frames. Four of them are non-temporal multimodal neural baselines that process a single frame (either the first frame or the last one) along with the question. Finally, the remaining five models are video question answering models, including the recently proposed methods, which process the entire video sequence in providing an answer to a given question.

**Most Frequent Answer** baseline (MFA) employs a simple heuristics and answers all the questions by using the most frequent answer in the training split. We use this simple baseline as a sanity check to inspect question biases. **Answer Type based Most Frequent Answer** model (AT-MFA) is a heuristics-based baseline like the MFA model. For each question querying a specific answer type (e.g. color, shape, boolean), it gives the same answer which corresponds to the most frequent answer observed for that answer type in the training split. In addition, **Random** model uniformly samples a random answer from the full answer space, whereas **Answer Type Based Random** model (AT-Random) makes random guesses based on the answer type (e.g. color, shape, boolean).

**LSTM** model is our third image-blind baseline that processes the question with an Long Short-term Memory network (LSTM) [39], and then predicts an answer to a given question ignoring the visual input. It encodes the question by using 256 hidden units and initializing word embeddings randomly.

Each question is represented with the last hidden state of the network by processing each individual input word sequentially.

**LSTM-CNN** baseline integrates both visual and textual cues by extending the LSTM model to additionally consider the features extracted from the 4-th convolutional layer of a pretrained ResNet-18 model. We evaluate both (non-temporal) single frame and video versions. In the former, each video is encoded with ResNet-18 model by taking into account either the first frame or the last frame, which are referred to as **LSTM-CNN-F** and **LSTM-CNN-L**, respectively. The video version, which we call **LSTM-CNN-V**, processes downsampled videos by using R3D [40], a 3-dimensional variation of ResNet-18, as visual feature extractor. All these three baselines concatenate the extracted visual and textual features to obtain a combined representation of the video and the question pair, feeding it to a multilayer perceptron network (MLP) which consists of 2 layers with unit size of 256 and with ReLU non-linearity. Finally, a linear layer generates scores for the answers. A dropout with a probability of 0.2 is used for both visual and textual representations.

**Memory, Attention, and Composition (MAC)** model [41] is a state-of-the-art compositional visual reasoning model. It decomposes the reasoning task into a series of attention-guided processing steps by isolating memory and control functions from each other. The attention mechanism considers visual and textual features jointly, which leads to robust encodings of the question and the image. Similar to the LSTM-CNN baseline, we have implemented two alternative versions. While the first one, which we name **MAC-F**, looks at only the first frame, the latter is called **MAC-L** and only pays attention to the last frame. Differently from the original MAC architecture, we use 256 units for control, read and write cells of MAC, insert batch normalization layers after convolutional layers, and apply dropout with 0.2 probability similar to the other baselines. We opted out self attention and memory gate in the write unit since they are optional.

**MAC-V** baseline extends the MAC model by considering the video frames sampled from the given video as the visual input. Like LSTM-CNN-V model, MAC-V also processes videos by using R3D. Unlike its non-temporal variations, MAC-F and MAC-L, where the read unit originally has spatial attention over the image, this temporal variation has a read unit that applies spatio-temporal attention over the entire video features extracted by R3D. MAC-V has same hyperparameters with MAC-F and MAC-L.

**TVQA** is a multi-stream state-of-the-art video question answering neural model [15]. To adapt this model to our dataset, we only use its video stream branch and omit the answer input by generating scores for the entire answer vocabulary. In parallel with other baselines, TVQA model also extracts visual features by using ResNet-18 architecture. Different from the original implementation, our TVQA implementation uses LSTM networks with 256 units, uses a MLP network with 2 layers. Unlike the original model, we do not use GloVe word embeddings [42] to make a fair comparison with the remaining baseline models.

**TVQA+** is another multi-stream video question answering model, which is built upon TVQA model. In contrast to TVQA, TVQA+ uses convolutional networks as sequence encoder instead of LSTM networks, replaces GloVe word embeddings with BERT embeddings [43], and implements a span proposal / prediction mechanism. We do not implement span proposal mechanism, and omit using BERT embeddings to compare TVQA+ with others more fairly as we disable GloVe embeddings in TVQA. Our TVQA+ implementation uses 256 hidden units in all submodules throughout the network, and it generates answer scores by feeding weighted average of fused multi-modal simulation-question representation into a linear layer.

**G-SWM** is an object-centric model [44], which is originally designed for simulating possible futures in a scene consisting of multiple dynamic objects. It models each frame in a video by two different latent variables encoding object and context features. We modify G-SWM to solve the reasoning tasks in CRAFT. In particular, our version of G-SWM takes in video frames resized to $64 \times 64$ pixels and extracts an object-centric representation of the input video thorough object and context features. These latent codes are then combined and concatenated with the LSTM-based question representation, similar to LSTM-CNN model, just before the final classifier layer.

**Implementation and Training Details**. Unless otherwise speficied, all learnable baselines are trained with Adam optimizer [45] with default hyperparameters. LSTM and single frame models are trained for 75 epochs with batch size of 64. All temporal baselines are trained for 30 epochs with batch size of 32. G-SWM is trained for 100 epochs using a batch size of 64 with Adam optimizer

Table 1: Performances of the tested baselines on the test set of the CRAFT dataset on easy and hard splits. C, CF, D, E and P columns stand for *Cause*, *Counterfactual*, *Descriptive*, *Enable* and *Prevent* tasks, respectively.

| | Baseline | Easy Setting | | | | | | Hard Setting | | | | | |
|---|---|---|---|---|---|---|---|---|---|---|---|---|---|
| | | C | CF | D | E | P | All | C | CF | D | E | P | All |
| Text only | Random | 7.41 | 5.25 | 5.09 | 4.72 | 5.76 | 5.24 | 7.52 | 4.62 | 5.08 | 3.99 | 5.73 | 4.98 |
| | AT-Random | 38.68 | 44.34 | 33.95 | 37.13 | 33.87 | 37.47 | 36.27 | 46.06 | 34.16 | 34.44 | 31.08 | 37.52 |
| | MFA | 34.16 | 43.28 | 23.53 | 33.79 | 29.72 | 30.72 | 32.03 | 43.94 | 23.20 | 30.78 | 28.02 | 29.98 |
| | AT-MFA | 46.50 | 47.21 | 37.57 | 51.87 | 50.46 | 42.03 | 49.67 | 47.17 | 36.55 | 49.08 | 49.28 | 41.12 |
| | LSTM | 49.18 | 53.14 | 38.29 | 53.63 | 56.68 | 44.69 | 49.69 | 56.24 | 37.25 | 55.91 | 50.10 | 44.52 |
| Single frame | LSTM-CNN-F | 50.21 | 55.23 | 44.86 | 55.60 | 53.46 | 49.07 | 46.08 | 48.12 | 35.54 | 47.25 | 50.31 | 40.64 |
| | LSTM-CNN-L | 52.06 | 55.63 | 43.12 | 55.60 | **57.14** | 48.42 | 50.33 | 54.44 | 38.88 | 51.25 | 47.85 | 44.66 |
| | MAC-F | 51.03 | 52.88 | 44.40 | 54.22 | 54.38 | 48.10 | 51.31 | 53.50 | 42.12 | 52.08 | 51.94 | 46.55 |
| | MAC-L | 45.88 | 53.08 | 44.54 | 54.03 | 49.77 | 47.83 | 45.10 | 53.80 | 41.46 | 50.25 | **53.37** | 46.05 |
| Video | LSTM-CNN-V | 51.03 | **61.42** | **48.12** | 56.58 | 56.45 | **53.01** | 48.69 | 54.89 | 41.36 | **52.58** | 52.97 | 46.50 |
| | MAC-V | **54.73** | 57.72 | 44.41 | 53.05 | 54.15 | 49.74 | 49.67 | 54.71 | **42.94** | 52.08 | 51.12 | **47.31** |
| | TVQA | 51.85 | 55.57 | 36.89 | 54.42 | 54.84 | 44.71 | 52.61 | **55.12** | 36.31 | 50.08 | 51.12 | 43.46 |
| | TVQA+ | 54.32 | 60.02 | 40.22 | **58.35** | 51.38 | 48.11 | **54.90** | **55.12** | 39.09 | 51.41 | 48.06 | 45.12 |
| | G-SWM | 51.03 | 55.29 | 37.05 | 55.60 | 53.92 | 44.69 | 51.96 | 48.68 | 37.77 | 49.42 | 52.35 | 42.47 |

| | | C | CF | D | E | P | All |
|---|---|---|---|---|---|---|---|
| | Human | 83.00 | 77.10 | 86.96 | 72.36 | 79.71 | 80.37 |

and a learning rate of 0.0001. Input videos are downsampled at 5 frame per second (fps) and their frames are resized to $112 \times 112$ pixels. We used mixed precision strategy to train baselines more efficiently on Tesla V100 and Tesla P4 GPUs, with the exception of TVQA+ which is trained by using full precision. Training single frame models take 2 minutes, and training video models take 20-30 minutes per epoch approximately. All word embeddings have the length of 256 and are randomly initialized. Pretrained convolutional video and image encoders are jointly trained with the rest of the networks. We use negative log-likelihood loss function for all models where the modelds predict a distribution over the set of possible answers. All models are tuned based on their performances on the validation split.

## 4.2 Results

In Table 1, we present the performances of the baseline models for each question type, considering both the easy and the hard settings explained in Section 3. We evaluate the performance of each model by comparing the answer token predicted by the model to the ground-truth and estimating the average accuracy accordingly.

Among the evaluated baselines, the text only models perform the worst, as expected, since they completely ignore the visual information present in the videos. Also, the performances of the single frame methods are typically worse than those of the video models, showing the importance of the temporal aspect of the questions that a single snapshot of the simulation does not carry enough information. Clearly, to excel in this task, a model must capture the interactions between the dynamic objects with each other and with the environment.

Moreover, as evident from the results of Table 1, there exists a substantial gap between the model performances in the easy and hard settings of CRAFT. Not surprisingly, this is not the case for the text-based baselines, in which it is not important whether a scene layout has been seen before during training or not. Overall, these results suggest that our tested multimodal methods are not able to generalize well to previously unseen scenes. They simply cannot fully recognize the physical interactions and corresponding events taking place in a video.

It is worth mentioning that the performances of the models vary between different question types in CRAFT. Out of the five question types, the models consistently perform poorly on the Descriptive questions in that the accuracies are around 23.5%-44.9% in the easy setting and 23.2%-42.9% in the

hard setting. The reason behind this could be attributed to the variety of the answers in this task as it includes questions covering both count, shape, and color of the object(s) (see Figure 4). On the other hand, the accuracies of the models on the remaining questions types are between 29.7% and 57.1% in the easy setting, and 28.0% and 56.2% in the hard setting.

LSTM-CNN-V baseline does reasonably well on the easy setting, but its generalization capability on the hard setting is not that good. TVQA performs worse than the LSTM-CNN-V baseline, which points out the fact that it is more tailor-fit to video question answering about TV clips, and its performance degrades when it does not have access to subtitles or the related concept detectors. Notably, MAC variants perform the best in the hard setting. MAC model, together with G-SWM, is a more expressive model specifically designed for compositional visual reasoning. G-SWM, however performs poorly in our experiments, which might be because the scenes in CRAFT usually consists of many objects, thus making it harder to learn decomposing a given video into objects and background. This problem might be alleviated by switching into a two-stage framework, in which G-SWM is pretrained first to improve its decomposition ability. For now, we left this as future work. Overall, the accuracies are not very high, indicating the shortcomings of the existing models in understanding physical reasoning.

In order to support our thesis stating that CRAFT is designed to be easy for humans, but difficult for machines, we also conducted a small human study. We asked 481 randomly selected CRAFT questions to 101 adults. We divided the questions into 5 parts with counterbalancing and every participant took one of the parts randomly. As well as answering the questions, the participants were allowed to state that the question was not clear enough to understand. Among these 94 participants, we only considered the ones who responded at least 75% of the questions , which corresponds to 56 people.As can be seen from Table 1, there is a large gap ($> 40\%$) between human subjects and neural baselines in the hard setting. However, we should say that humans had more difficulty answering Enable questions, but even for that question type the gap is big ($> 20\%$). We must admit that detailed studies on human subjects solving CRAFT tasks are also required to better understand differences between humans and machines.

# 5    Conclusion

We have presented CRAFT, a new video question answering benchmark to challenge intuitive physics capabilities of the current machine learning algorithms. Motivated by the theories of force dynamics in cognitive linguistics, CRAFT requires models to perform temporal and causal reasoning and even to imagine alternative versions of the events occurring in videos. Our results demonstrate that, while reasoning about the physical interactions between objects seem intuitive to humans, these questions cannot be solved reliably by the current state-of-the-art models. At present, there is large room for improvement when compared to human performance. In our experiments, we did not report the results of recent neuro-symbolic models (e.g. Neuro-Symbolic Dynamic Reasoning (NS-DR) [18]). Such approaches are very interesting and worth pursuing, but they currently require extra object-level annotations. Another exciting direction is to test other object-centric models like G-SWM. However, it seems that they might require extra pretraining or self-supervised objectives, as explored by [46].

Current version of CRAFT includes multiple patients in cause, enable, and prevent tasks, but does not include multiple affectors. Hence, it might be possible to extend CRAFT with these kind of more complex object relationships. Moreover, new object attributes, such as density, can be integrated using material textures. Finally, the programs designed for our tasks depend on the end results of the simulations to be able to provide correct answers to the questions. Investigating temporally local relationships between objects might be interesting as well. We believe that developing more effective algorithms for solving CRAFT tasks is an exciting research direction for artificial intelligence systems mimicking humans for causal reasoning about forces and interactions.

# Acknowledgments

This work was supported in part by GEBIP 2018 Award of the Turkish Academy of Sciences to E. Erdem and T. Goksun, BAGEP 2021 Award of the Science Academy to A. Erdem, and AI Fellowship to Ilker Kesen provided by the KUIS AI Center.

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
