# OpenReview forum: "CRAFT: A Benchmark for Causal Reasoning About Forces and inTeractions"
_NeurIPS.cc/2021/Track/Datasets_and_Benchmarks/Round1 — Submitted to NeurIPS 2021 Datasets and Benchmarks Track (Round 1)_

### Official Review · Reviewer_p1Xy · 2021-07-04

**Rating:** 6
**Confidence:** 3
**Correctness:** Correct to the best of my knowledge.
**Clarity:** Clarity can be improved.

**Strengths:**

The paper proposes an interesting task that (under appropriate parameter initializations) is easy for humans via "commonsense reasoning" but difficult for machine learning. The textual questions are about objects in the scene in different arrangements. The difficulty of the task incorporates text parsing, visual decoding, and learning effective physics simulations.

**Weaknesses:**

Figure 2 shows that the size of the dataset is generated in part by perturbing within small bound shte location of objects in fixed configurations. On the other hand, the paper also notes that “Humans do not accurately reason about the objects for some counterfactual cases as subtle changes in the scenes result in very different outcomes.”, presumably because of predicting events such as collisions between moving objects that can be quite sensitive to absolute position information.

It is not exactly well-argued by the paper why the visual question-answering task corresponds necessarily to “human commonsense reasoning”. The task itself has the structure of visual synonyms layered on top of a physics engine --- if a neural network appropriately parses the sentence structure (even with verbal variation) to an appropriate mapping in the physics engine, and had access to a physics engine (let’s say), then it could sufficiently “solve” the task. As to the question of whether a neural network can “learn” a physics engine, it’s not necessarily well argued why this is a good benchmark: if we really wanted to solve physics problems with complex sensing, we should endow the machine learning pipeline with the physics engine.
Overall, since ultimately the dataset is generated from an underlying physics engine, it’s not clear to me that the difficulty introduced by the dataset maps well onto the space of machine learning, in the sense that the difficulty seems a bit artificial in the sense that it is difficult for neural networks (for example) to learn physics engines. The paper does not provide evidence, however, that a suitably engineered approach to visual question answering (decoding the vision state by CNN and passing scene reconstruction into a physics engine) would solve the task. I surmise that if one really wanted to solve the task, that is what one would do instead.


**Additional Feedback:**

Perhaps framing the paper more narrowly could help the sharpness of the claims, eg not emphasizing "commonsense reasoning" (since after all, the "commonsense reasoning" is more about "intuitive physics", e.g. that humans can think as a worse version of an actual physics engine.

**Documentation:**

Documentation appears reasonable.

**Relation To Prior Work:**

Relation to prior work and other types of similar benchmarks could be improved.

**Summary And Contributions:**

The paper proposes a new visual question answering task which requires understanding complex physical reasoning, in the sense of "decoding visual data" to answer questions about physical simulations under alternative configurations of the data. The dataset emphasizes visual question-answering about “cause, enable, and prevent” and claims this is related to “commonsense knowledge”

===
Thanks for the response!
I have raised my score a bit but would defer the substantive questions of intuitive physics engines as benchmarks to other reviews/the AC.

---

> ### Author Response · Authors · 2021-07-12
> **Thank you for your valuable feedback, we have carefully addressed your concerns below.**
>
> Thank you for your insightful comments and valuable feedback. Below we address your concerns and questions:
>
> **[On the perturbations]** In our work, we consider performing small perturbations in both scene construction and question generation steps. That said, the reasons behind these perturbations and the kinds of perturbed scene elements are totally different. In the scene construction, we randomly perturb the static scene elements (ramp, platform, button, basket, etc.) to increase the diversity within each scene template. On the other hand, in the question generation, we apply minor random perturbations to each dynamic object in a video (cube, triangle or circle) to eliminate noisy and ambiguous inputs for the reason that some psychological studies, e.g., (Kubricht et al., 2017) show that humans’ predictions and judgements about physical activities are affected under these conditions. In particular, the random perturbations in this step involve changing the speed of the dynamic objects. For a given question, we expect to obtain the same answer in case we increase or decrease the object’s speed by 2%, and use this strategy as a filtering scheme to remove such kinds of possibly ambiguous videos.
>
> James R. Kubricht, Keith J. Holyoak and Hongjing Lu. Intuitive Physics: Current Research and Controversies. Trends in Cognitive Sciences, Vol. 21, October 2017, pages 749-759.
>
> **[On commonsense reasoning and the use of physics engine]** We respectfully disagree with the reviewer. In psychology and cognitive sciences, synthetic scenes generated either manually or by a physics engine are commonly used to assess the predictions and the judgments of the humans regarding intuitive physics or even intuitive psychology, which are the core components of human commonsense reasoning. Moreover, researchers argue that intuitive physical inference in humans might be based on a mental physics engine that resembles physics engines we use in computers (Ullman et al., 2017). In recent years, similar efforts have been observed in the machine learning community as mentioned in the “Intuitive Physics in Artificial Intelligence” subsection of the Related Work section of our paper. The prior work TIWIQ [23], CLEVRER [18], and CLEVR_HYP [24], which are the most similar to our work, all include scenes synthetically generated by a physics engine. We simply follow this long and established tradition. However, as compared to these studies that involve moving objects interacting with each other, our work includes more complex interactions. That is, in CRAFT, a moving object might both interact with static scene elements as well as other dynamic objects, hence the questions require understanding both object-scene and object-object interactions.
>
> Tomer D. Ullman, Elizabeth Spelke, Peter Battaglia, and Joshua B. Tenenbaum. Mind Games: Game Engines as an Architecture for Intuitive Physics. Trends in Cognitive Sciences, Vol. 21, Issue 9, September 2017, pages 649-665.
>
> **[On the relation to prior work]** We would be very glad if the reviewer points out the related work that we missed which should be cited. We are happy to include and discuss them in our paper.

---

### Official Review · Reviewer_Bpwc · 2021-07-05
**Valuable dataset for physical causal reasoning. Some additional baseline performance measures required.**

**Rating:** 8
**Confidence:** 2

**Strengths:**

The construction of scenes and the corresponding question-answer pairs are usefully grounded in cognitive and linguistic theories that attempt to account for humans' physical reasoning capabilities. The construction of the dataset is clear and well principled. I am not familiar with the related work, so it is difficult for me to evaluate the novelty, but based on their description I understand this to be a useful and novel extension to existing VQA datasets.

**Weaknesses:**

This seems like a self-evidently valuable type of dataset to construct. I have two concerns with this work that I hope the authors can address.

First, insofar as this dataset is intended to measure advances in AI's ability to mimic human intuitive physics, it seems vital to obtain more robust measurements of how well humans can reason about these scenes. The baseline study that they conducted is useful and informative, but it's small (481 questions, 101 adults). In addition to measuring individuals' accuracy, it seems important to measure inter-annotator reliability. Perhaps these measures wouldn't be needed if the dataset were intended to be prescriptive with respect to physical causal reasoning, but the cognitive theory behind it suggests that it's intended to be descriptive of human reasoning processes.

Second, I have some concerns with the language in the question-answer pairs. (For context, I'm a native English speaker.) (1) The "do" construction in English signals a "Yes"/"No" question. It sounds odd to respond "True"/"False" to these questions. (2) I'm not surprised that annotators had more difficulty with Enable questions than with the other categories of questions, because in the context of questions like this it seems that "enable" could signal either active facilitation or a lack of interference. The description in the second paragraph of the Question Types section suggests that the first meaning is intended, but I think the word "enable" is ambiguous. I think that I myself wouldn't know how to answer without further instructions about which of these meanings is intended. If the goal is to measure humans' intuitive physics, then some clarification is probably required for respondents. That probably goes for the other categories as well.

**Additional Feedback:**

No additional comments.

**Clarity:**

The paper is generally clearly written. There are a couple points that I think need clarification. I can guess at most of the answers, but it would be helpful to reword or briefly clarify these.
- What exactly does it mean that "the language component is largely missing" in previous work?
- What does it mean that previous datasets "lack visual variations"?
- What does it mean that CRAFT questions "require multiple counterfactual simulations to be explored"?
- Why is it "crucial to have language variety"?

More substantially, I don't understand the examples in Figure 1. For instance, the first question-answer pair says that 2 objects fall to the ground. Looking at the scene, I would have thought the answer was 4. I'm also not sure how I would go about answering the other four categories of questions. How can I tell that the small brown sphere causes the tiny yellow box to enter the basket? Perhaps these categories can't be meaningfully represented without animation, but this needs to be clarified.

**Correctness:**

The claims are correct to the best of my knowledge. The dataset seems to be well motivated and soundly constructed, and the evaluations seem appropriate and robust.

**Documentation:**

This is all provided in a nice clear format.

**Ethics:**

No.

**Relation To Prior Work:**

This is generally well described, though there are a couple points that merit clarification as described above.

**Summary And Contributions:**

The authors develop a dataset for Visual Question Answering (VQA) that is motivated by "intuitive physics": humans' natural ability to reason causally about physical forces and events. They evaluate a number of baseline and state-of-the-art methods for question answering and show that these methods do not perform well on this dataset relative to human reasoning abilities. This suggests that their dataset will be useful for detecting and quantifying advances in this domain.

If I understand the Related Work section correctly, this represents a valuable contribution toward the development of AI that is able to mimic humans' physical reasoning abilities. The context vis-a-vis previous datasets merits some clarification, and I have some concerns about the language used in the dataset and about the need for additional quality control via human performance baselines.

---

> ### Author Response · Authors · 2021-07-12
> **Thank you for your valuable feedback, we have carefully addressed your concerns below.**
>
> Thank you for your insightful comments and valuable feedback. Below we address your concerns and questions:
>
> **[On inter-annotator agreement]** Inter-annotator agreement might lead to false interpretations for our study. We asked participants questions with multiple choices. Therefore, they didn't annotate qualitatively (following a pre-written guideline, see Artstein, 2017), but they only selected the answer based on their common sense. However, following the reviewer's suggestion, we also measured inter-annotator agreement with Fleiss kappa. Since cognitive load can be a confounder, we divided participants into 5 parts for the human subjects study. They solved approximately 20 questions equally distributed by the categories (descriptive, cause, enable, prevent, counterfactual). We measured the inter-annotator agreement of each part, and our results revealed that Fleiss' kappa values are between 0.26 and 0.56, p<.001, showing a fair to moderate agreement between the participants (Viera & Garret, 2005).
>
> Ron Artstein. Inter-annotator Agreement. In the Handbook of Linguistic Annotation, edited by Nancy Ide and James Pustejovsky, pages 297–313. Springer, Dordrecht, 2017.
>
> Anthony J. Viera and Joanne M. Garrett. Understanding interobserver agreement: the kappa statistic. Family Medicine, Vol. 37, No. 5, May 2005, pages 360-363.
>
> **[On the language used in the question-answer pairs]** We asked native English speakers to check the questions to eliminate any ambiguity and/or grammatical errors. Humans indeed answered the questions that start with “Do” as either Yes or No. The confusion arises from the typo that we mention them as True/False. We apologize for this mistake and we will correct this in the revised version of the paper.
>
> As for different question types, our results are compatible with the literature on force dynamics that understanding some causal verbs is simpler than the others (Wolff and Barbey, 2015). Enable is not as simple as Cause. In Enable, the second force seems to be additional. It involves more than two forces different from Cause. However, different from Prevent, Enable does not involve two opposing forces. Developmental research suggests that children conceptualize these categories by 5 years of age. However, Enable and Prevent are more difficult for 3 and 4 year olds (Göksun et al., 2013; George et al., 2019). Intuitive physics studies with children did not provide clarifications or instructions during testing on these categories, as these are supposed to be “intuitive”. Following those we did not include additional instructions and explanations for human study.
>
> T. Göksun, N.R. George, K. Hirsh‐Pasek, R.M. Golinkoff. Forces and motion: How young children understand causal events.
> Child development, Vol. 84, Number 4, 2013, pages 1285-1295.
>
> N.R. George, T. Göksun, K. Hirsh-Pasek, R.M. Golinkoff. Any way the wind blows: children’s inferences about force and motion events. Journal of experimental child psychology, Vol. 177, 2019, pages 119-131.
>
> **[On the clarity of the paper]** The majority of the prior work on intuitive physics has explored learning physics by only watching dynamic scenes where the studied tasks involve predicting the future or placing a certain object to a certain location in order to achieve a well-defined task. TIWIQ, CLEVRER and CLEVR_HYP datasets are exceptions, which are visual question answering benchmarks like our proposed CRAFT dataset, but the scenes exist in those datasets are derived from the same environment and the interactions with the static elements of the scenes are in general missing. Hence, CRAFT requires understanding language as well as physics for more complex scenes. Moreover, answering some questions, e.g. "Does the small gray box enter the basket, if any other single one of the objects is removed?" asks models to reason about multiple possible simulations for each other object in order to provide the correct answer. In automatically generating the questions, we utilize carefully designed templates, some of which are paraphrases of each other. In our work, we inherit the design principles used in CLEVR and its extensions, including CLEVRER and CLEVR_HYP, which are found to be effective in increasing the dataset size as well as helping models to avoid overfitting to the question texts.

---

> > ### Comment · Reviewer_Bpwc · 2021-07-21
> > **I still think more annotations are required...**
> >
> > Thanks to the authors for their thoughtful responses and for clarifying certain points of confusion.
> >
> > I remain uncertain about whether this dataset should be understood as reflecting (1) prescriptive truth about causal relationships and events, or (2) humans' intuitive physics. As in, should this dataset be used to test whether computers can understand complex physical causal relationships, or to test whether computers can learn to mimic humans' intuition about these types of relationships?
> >
> > If (1), then it seems that human annotations are largely irrelevant. All we should care about is whether machines can achieve the right answers, where "right" is a prescriptive notion that is intrinsic to each scene. If (2), then what we care about is whether computers can mimic human behavior, in which case it seems vital to have a large number of human annotations to form the ground truth. Also if (2), then it seems vital to distinguish the linguistic and cognitive levels. There's linguistic ambiguity in the word "enable," between something like active facilitation vs. lack of interference, but this doesn't necessarily mean that there's a corresponding cognitive ambiguity. I suspect that if annotators were told that "enable" referred to the former meaning, they could achieve much higher accuracy and agreement. Speaking of agreement, I don't understand the remark that "Inter-annotator agreement might lead to false interpretations."
> >
> > Overall, I think this dataset is very useful, but some more care needs to be given to these distinctions.

---

### Official Review · Reviewer_aB2M · 2021-07-07
**Novel dataset but with some bias and quality issue**

**Rating:** 5
**Confidence:** 4

**Strengths:**

The main contribution of this paper is the creation of a new visual question answering that requires causal reasoning about physical forces and object interactions. This dataset differs from previous work [23, 18, 24] as it contains more complex scenes, i.e., the moving objects can interact with both static and dynamic scene elements while in [23, 18, 24], they can only interact with dynamic scene elements. I believe the move from simple environment to a more complex one is a reasonable and natural next step for the community. The dataset, if of good quality, could be useful for the research community.

**Weaknesses:**

- The fact that `Text only` models perform so well (e.g. in Table 1, especially under hard setting, the LSTM model is better than or comparable to a few single frame and video models) might suggest there is a certain degree of language bias in the dataset, i.e., the questions and answers are correlated. While the authors put some efforts on balancing the answer distribution (L206), this correlation is not studied. It would be good to identify and eliminate these biases before releasing the dataset.

- Follow the last point, I am also curious how the models perform if using stronger language features/embeddings like GloVe and BERT. I feel the text only models could be further improved using these features/embeddings.

- Data quality. See Correctness section where I pointed out potential data quality issue.

**Additional Feedback:**

- I noticed in L108 the train/val/test splits in the hard setting is based on different scenes? Could you confirm that different `scenes` means different `environments``?
- I don't quite see a difference between `Cause` and `Enable` question types. Maybe they should be merged? Otherwise, please distinguish them clearly.
- Could you explain why TVQA+ is trained using full precision? It should be good to add one more comment explaining this in the paper.


---------

-- after discussion, final review (copied from my reply)


Thanks for responding to my question and concerns. I am convinced the language bias is not an issue for this work. However, there is one question that is not answered or not answered clearly. In `Additional Feedback`: `I don't quite see a difference between Cause and Enable question types. Maybe they should be merged? Otherwise, please distinguish them clearly.` I am aware that the authors responded to a similar concern from Reviewer `Bpwc`, but that still looks confusing to me. For example, in Figure 1, for the 1st `Enable` question, I can replace the word `enable` with `cause` and the question would still be valid, in that case, should it be categorized as a `Cause` question? Likewise, I can also replace the word `cause` with `enable` for the 1st `Cause` question, and then it might be categorized as an `Enable` question. With this confusing categorization, I don't think the current evaluation of these different categories is meaningful. For this reason, I'll keep my original rating, and encourage the author to address this in the 2nd round.

BTW, for the negative value with high magnitude, you can probably replace it with a smaller value then fp16 should work well. I.e., replace `1e10` with `1e5`.

**Clarity:**

Some parts can be further improved:
- Figure 1: Please add a legend for bucket as well. I was initially confused that the whole figure represents the bucket, as it looks the same as the smaller one.
- Figure 1: Showing the whole video in a single frame looks nice and succinct, but it lacks a bit clarity: I cannot tell whether the small brown circle collided with the small yellow square or not. Thus cannot answer the 1st Enable question. Similar for the 2nd Enable question. I recommend the authors to represent it as a sequence of frames like Figure 3.
- In L 346, could you detail why only consider the ones who responded at least 75% of the questions? How many questions are left after this filtering? If the pool is too small, the human performance might not be accurate.

**Correctness:**

- The answer to the 2nd Counterfactual question in Figure 1 does not look correct to me. For example, if the large cyan triangle is removed, the small gray box may still end up in the platform (where the large cyan triangle initially sets.). To me, I would answer "Not sure, removing large cyan triangle increases the chance that the small gray box enter the basket, but this chance is not 100% since the small gray box may stuck in the platform."
- Following the question above, I am a bit worried about the overall human performance of this dataset. I am aware that the authors performed a filtering process for the final dataset in L163, but this filtering is not described in detail. The relatively low human performance in Table 1 also makes me wonder the potential quality issue of the data.
- The paragraph starts from L310 makes a few false claims about Table 1, e.g., "Among the evaluated baselines, the text only models perform the worst,". As noted in the Weaknesses section, the best text only model `LSTM` is better than one Single frame model and two video models under hard setting.

**Documentation:**

I checked the supplementary file, website, and code, the documents are of reasonably good quality to support reproducibility.

**Ethics:**

I did not see any ethic issue with this dataset.

**Relation To Prior Work:**

There are 3 prior work TIWIQ [23], CLEVRER [18], and CLEVR_HYP [24] that are most similar to this work. Compared to all the 3 work, this paper is unique as it involves more complex scenes (L71). By compare the paper and the 3 references in detail, I notice the main difference lies in that this work contain both static scene elements and dynamic scene elements to interact that a moving object can interact with, while in the 3 references, the moving object can only interact with other dynamic elements. I believe this part can be better explained and added to the related work section — so readers can quick get a sense of where the difference is, it is rather hard to get it from a `explained later` sentence, "however the scenes in our benchmark are more complex, as explained later." (L71).

**Summary And Contributions:**

This work presents an interesting new dataset supporting the research on visual question answering that requires causal reasoning about physical forces and object interactions. Compared to previous work [23, 18, 24], the proposed dataset contains more complex scenes. The data is generated automatically from virtual scenes. 3 sets of baselines are established for the newly created data. However, there are several major weaknesses, mostly regarding the baselines and the data bias and quality.

---

> ### Author Response · Authors · 2021-07-12
> **Thank you for your valuable feedback, we have carefully addressed your concerns below.**
>
> Thank you for your insightful comments and valuable feedback. Below we address your concerns and questions:
>
> **[On data bias and text-only models]** We agree that text-only models seem to have unexpectedly good results as compared to other baselines.  We think the underlying reason behind this is not the language bias found in our dataset. As pointed out, we explicitly followed certain post-processing procedures to eliminate such kinds of biases. To further evaluate the correlation between the questions and the answers, we have implemented a new bag-of-words based QA model with a naive Bayes classifier. It respectively achieves 44.86% and 43.34% overall accuracy on the easy and hard settings, indicating that questions and answers are not highly correlated. As suggested, we have implemented different versions of our text-only LSTM model by using GloVe and BERT embeddings and observed that switching to them does not introduce any substantial performance gain. GloVe-based model gives 34.53% and 33.61% overall accuracy while the BERT-based model performs much better with accuracies of 42.90% and 42.52% in the easy and hard settings, respectively. Still, they are worse than the LSTM model with randomly initialized text embeddings. Detailed results are given in the tables below.
>
> **Easy Setting**
>
> |Baseline|C|CF|D|E|P|All|
> |----------|:-------:|:-------:|:-------:|:-------:|:-------:|:-------:|
> |GloVe|35.39|43.22|30.35|32.22|28.80|34.53|
> |BERT|46.91|50.59|37.55|47.54|51.15|42.90|
> |BoW (3-gram)|51.23|52.77|39.65|51.67|44.01|44.86|
> |LSTM|49.18|53.14|38.29|53.63|56.68|44.69|
>
> **Hard Setting**
>
> |Baseline|C|CF|D|E|P|All|
> |----------|:-------:|:-------:|:-------:|:-------:|:-------:|:-------:|
> |GloVe|29.08|44.17|29.39|29.62|28.43|33.61|
> |BERT|51.63|52.12|36.52|47.25|50.31|42.52|
> |BoW (3-gram)|54.25|53.53|37.04|51.25|46.22|43.34|
> |LSTM|49.69|56.24|37.25|55.91|50.10|44.52|
>
> **[On the data quality and human performance]**
> We are aware that presenting dynamic content in a PDF file is challenging, thus we refer the readers to our project website to explore sample videos from our dataset. From the overlaid images, one cannot easily perceive the speed of the objects, which might lead the reviewer to make some false judgments about the 2nd counterfactual question in Figure 1. To eliminate such misinterpretations, we will include a link to the original video. For your view, it can be viewed from https://streamable.com/e/vftsno
>
> We respectfully disagree that humans have low performance on CRAFT. We think otherwise – considering the complexity of the scenes, humans indeed achieve pretty good results in answering the questions in our dataset. As reported in prior psychological studies (Kubricht et al., 2017), humans might be affected by common misconceptions and biases when predicting and judging some physical events, especially the ones giving ambiguous perceptual inputs. Hence, in our work, to remove such possibly ambiguous videos, we apply minor random perturbations to each dynamic object in a video to verify whether the same answer is obtained for all such cases, and exclude those questions that do not pass this verification step. These perturbations involve changing the speed of the objects. For a given question, we expect to obtain the same answer whether we increase or decrease the object’s speed by 2%. We will discuss these details thoroughly.
>
> Kubricht et al. Intuitive Physics: Current Research and Controversies. Trends Cogn. Sci., Vol. 21, 2017.
>
> **[On clarity and relation to prior]** We apologize for the unclear parts in Figure 1. We will correct these. In regard to our human evaluation, we actually shared our study's link with 94 participants, but 34 of them left the study without answering the questions (or just answer the first few questions) and 4 of them reported having problems with distinguishing the colors. Therefore, only 56 of them fully answered the questions. We will revise the corresponding sentence as "Among these 94 participants, we only considered those who do not have difficulty in distinguishing between different shades of red, yellow and green colors and almost fully responded to the questions, which correspond to 56 people that answered at least 75% of the questions”.
>
> About the relation to prior work, as correctly summarized, our main difference lies in the characteristics of the videos in our dataset. In particular, each video involves interactions between a moving object and both static and dynamic scene elements. We will stress these differences more.
>
>
> **[On additional feedback]** (1) Different scenes in the Hard Setting mean different environments. (2) We trained TVQA+ model with full precision since the masking/padding operation in the structured attention module of the original implementation does not allow us to take advantage of mixed precision training. That is, padded values are represented with negative values with high magnitude (~1e10), which is not representable in half precision.

---

> > ### Comment · Reviewer_aB2M · 2021-07-17
> > **Reply to response**
> >
> > Thanks for responding to my question and concerns. I am convinced the language bias is not an issue for this work. However, there is one question that is not answered or not answered clearly. In `Additional Feedback`: `I don't quite see a difference between Cause and Enable question types. Maybe they should be merged? Otherwise, please distinguish them clearly.` I am aware that the authors responded to a similar concern from Reviewer `Bpwc`, but that still looks confusing to me. For example, in Figure 1, for the 1st `Enable` question, I can replace the word `enable` with `cause` and the question would still be valid, in that case, should it be categorized as a `Cause` question? Likewise, I can also replace the word `cause` with `enable` for the 1st `Cause` question, and then it might be categorized as an `Enable` question. With this confusing categorization, I don't think the current evaluation of these different categories is meaningful. For this reason, I'll keep my original rating, and encourage the author to address this in the 2nd round.
> >
> > BTW, for the negative value with high magnitude, you can probably replace it with a smaller value then fp16 should work well. I.e., replace `1e10` with `1e5`.

---

> > > ### Comment · Program_Chairs · 2021-07-20
> > > **Additional comment**
> > >
> > > Reviewer aB2M objects to our categorization in Table 1 in which we divide the questions in our CRAFT benchmark dataset according to the causal verbs commonly studied in the cognitive linguistics, in particular in Phillip Wolff's Force Dynamics Theory (i.e. Cause, Enable, Prevent). We are aware that for non-native English speakers or those who are not familiar with Force Dynamics Theory, this categorization may not be 100% intuitive. Table 2 (attached) from Ref [8] (Wolff, 2013) clearly puts the differences between the three verbs into correct context. In our response we tried to clarify this, but it seems this was not enough for Reviewer aB2M. We are happy to include a more detailed explanation in our revision, but I think that it is quite unfair not to increase the score based on this point — especially considering that one can easily merge the Cause and Enable categories in a single column (since we do not train separate models for each category), but this will clearly contradict with the literature in cognitive science. We were hoping to have more correspondences with the reviewers during the discussion period.

---

### Decision · Program_Chairs · 2021-07-27

**Decision:**

Reject

**Comment:**

Reviewers agree that this paper is proposing an interesting task of visual question answer tasks using complex physical reasoning. However, there are a few issues that we hope authors clarify to resubmit in the 2nd round. First issue around human performance (raised by all reviewers). For example, inter-annotator kappa coef seems very low (0.26-0.56). While author’s argue that the way study was conducted did not request folks to annotate qualitatively, it remains unclear what would it mean to have the human labels if humans can’t agree with each other. While authors argue that the performance itself is not bad given the high complexity of the task, the fact that humans don’t agree means that some (maybe most?) humans get the answer ‘wrong’ in some way. How do we reconcile this for the goal of this dataset? It would be very helpful to further clarify this point. Second, related to the first one, the final goal of this dataset could use more clarification. One reviewer put this elegantly, so I paste their words here "I remain uncertain about whether this dataset should be understood as reflecting (1) prescriptive truth about causal relationships and events, or (2) humans' intuitive physics. As in, should this dataset be used to test whether computers can understand complex physical causal relationships, or to test whether computers can learn to mimic humans' intuition about these types of relationships?”